# Geometric, Electronic, and Optoelectronic Properties of Carbon-Based Polynuclear C_3_O[C(CN)_2_]_2_M_3_ (where M = Li, Na, and K) Clusters: A DFT Study

**DOI:** 10.3390/molecules28041827

**Published:** 2023-02-15

**Authors:** Imene Bayach, Atazaz Ahsin, Safi Ullah Majid, Umer Rashid, Nadeem S. Sheikh, Khurshid Ayub

**Affiliations:** 1Department of Chemistry, College of Science, King Faisal University, Al-Ahsa 31982, Saudi Arabia; 2Department of Chemistry, Abbottabad Campus, COMSATS University Islamabad, Abbottabad 22060, KPK, Pakistan; 3Chemical Sciences, Faculty of Science, University Brunei Darussalam, Jalan Tungku Link, Gadong BE1410, Brunei

**Keywords:** polynuclear superalkali clusters, excess electrons, nonlinear optical response, dynamic hyperpolarizability

## Abstract

Carbon-based polynuclear clusters are designed and investigated for geometric, electronic, and nonlinear optical (NLO) properties at the CAM-B3LYP/6-311++G(d,p) level of theory. Significant binding energies per atom (ranging from −162.4 to −160.0 kcal mol^−1^) indicate excellent thermodynamic stabilities of these polynuclear clusters. The frontier molecular orbital (FMOs) analysis indicates excess electron nature of the clusters with low ionization potential, suggesting that they are alkali-like. The decreased energy gaps (E_H-L_) with increased alkali metals size revael the improved electrical conductivity (σ). The total density of state (TDOS) study reveals the alkali metals’ size-dependent electronic and conductive properties. The significant first and second hyperpolarizabilities are observed up to 5.78 × 10^3^ and 5.55 × 10^6^ au, respectively. The β_o_ response shows dependence on the size of alkali metals. Furthermore, the absorption study shows transparency of these clusters in the deep-UV, and absorptions are observed at longer wavelengths (redshifted). The optical gaps from TD-DFT are considerably smaller than those of HOMO-LUMO gaps. The significant scattering hyperpolarizability (β_HRS_) value (1.62 × 10^4^) is calculated for the **C3** cluster, where octupolar contribution to β_HRS_ is 92%. The dynamic first hyperpolarizability β(ω) is more pronounced for the EOPE effect at 532 nm, whereas SHG has notable values for second hyperpolarizability γ(ω).

## 1. Introduction

Nonlinear optical (NLO) materials are at the front line of research in interdisciplinary science and laser-based technology due to their fundamental applications in the field of optoelectronics [1,2,3,4]. Photonic devices, laser-based technology, endoscope, and sensors are examples of well known technologies where NLO materials have possible applications [5,6,7,8,9,10,11,12]. To design and synthesize the NLO materials, much efforts are exerted to understand the origin of nonlinearity in molecules and clusters in order to correlate NLO responses to electronic structure and molecular geometry. Polarization, asymmetric charge distribution, asymmetric crystal packing, and π-conjugated electron transport routes are all required for NLO materials. Because of their high thermal stability and transparency, inorganic nonlinear optical materials have been preferred over organic ones [13]. Some inorganic borates crystals, such as KB_5_ (KB_5_O_8_H_2_O), BBO (-BaB_2_O_4_), and LiB_3_O_5_ (LBO), have been investigated as good NLO materials, particularly in the ultraviolet range [13].

For obtaining high-performance NLO materials, several strategies were proposed, which include bond length alternation (BLA) [14], doping metal atoms [15], push–pull mechanisms from donor to acceptor [16], modification of sp^2^ hybridized carbon nanomaterials [17], designing octupolar molecules [18], multidecker sandwich complexes [19], and excess electron induction [20]. 

The introduction of excess electrons into molecules and clusters is the most viable technique to escalate hyperpolarizability. The availability of loosely bound electrons predominantly decreases the excitation energies for the crucial transition [21,22,23]. Excess electrons in molecules and crystals behave similarly to Rydberg orbitals, which are positioned outside the parent molecules and are held loosely [24,25,26]. Several studies have demonstrated the substantial role of the diffuse excess electrons in compounds for developing NLO materials. Wei Chen et al. investigated the Li@calix[4]pyrrole electride complex, which has a significant static hyperpolarizability (β_o_) value of up to 7.3 × 10^3^ au, where the presence of excess electrons has a significant role [27].

Theoretically designed compounds having excess electrons that are further classified into [28] alkalides [29], alkaline-earthides [30], metalides [31], and electrides [32]. Alkalides are complex compounds in which alkali metals bear the negative charge (Li^−^, Na^−^, K^−^) [33]. On the other hand, electride complexes have anionic sites occupied by the electron inside the complexes [34]. Furthermore, the alkaline–earthides were recently introduced to excess electrons compounds, where the alkaline earth metals hold a negative charge [35]. Interestingly, superalkali clusters are a new class of materials that can transport electrons, making them useful for the fabrication of electro-optical materials [36].

Superalkalis clusters with lower ionization energy (IE) than alkali metal elements are well known due to their powerful reducing capabilities. The very first report about superalkalis was obtained in 1982 by Gutsev and Boldyrev for Li_3_O, Li_2_F, and Li_4_N clusters [37]. These clusters with unique qualities, such as tuneability in their electrical properties and the ability to function as a bridge between micro and macro materials, are of great interest to cluster science. Recent advances in cluster science show the potential applications of superalkali clusters, i.e., the reductive materials, helium and hydrogen storage, catalysis, supersalt formation, and nonlinear optics [38,39,40,41]. 

Superalkali clusters are excellent candidates for creating optical and NLO materials because of their excellent tunable electronic and structural properties. The decreased excitation energy may be responsible for electrons shifting from HOMO to LUMO, as they are loosely bound. Based on the intriguing characteristics of superalkali clusters, these were used to fabricate NLO materials. In this regard, two-dimensional materials doped superalkali, and they play an essential role in triggering the hyperpolarizability response. Sun et al. theoretically designed superalkali-based alkalides Li_3_^+^(calix[4]pyrrole)M^−^, Li_3_O^+^(calix[4]pyrrole)M^−^, and M_3_O^+^(calix[4]pyrrole)K^−^ (M = Li, Na, and K), where the hyperpolarizability response is recorded up to 34 718 au [42]. Similarly, Faizan Ullah et al. reported a noticeable enhancement in the NLO response of the A1_12_P_12_ nanocluster by using Li_4_N, Li_2_F, and Li_3_O superalkalis as the source of the excess electrons [43]. Furthermore, macrocyclic oligofurans ring doped with superalkali clusters were also reported as a new kind of nonlinear optical material where a larger hyperpolarizability response is attributed to the presence of loosely bound electrons [44]. 

Although a larger number of superalkali clusters were theoretically designed, very limited studies have been conducted to show the possibility of using polynuclear superalkali (undoped) clusters as NLO materials. Srivastava et al. investigated the electronic and nonlinear optical properties Li_n_F (*n* = 2–5) and M2X small clusters as excess electron compounds where the β_o_ increases up to 10^5^ au for Li_2_F [45,46]. Our group investigated the static and dynamic hyperpolarizability response of M_2_OCN and M_2_NCO (M = Li, Na, K) superalkali clusters as excess electrons candidates where the second hyperpolarizability γ(ω) values were calculated up to 2.1 × 10^8^ au [45]. 

Superalkali clusters can be mononuclear, bimetallic, and polynuclear based on their rational design and elemental composition. We are interested to investigate carbon-based polynuclear clusters for electronic and NLO properties. These clusters are more stable than conventional mononuclear superalkali clusters and might possess better electronic and NLO properties. The previous development in the family of superalkalis and their tunable electronic properties prompted us to further investigate polynuclear clusters for optical and NLO properties. Polynuclear carbon-based clusters C_3_O[C(CN)_2_]_2_M_3_ (where M = Li, Na, and K) are investigated using DFT.

## 2. Results and Discussion

### 2.1. Optimized Geometries and Thermodynamic Stabilities

The optimized geometries of carbon-based polynuclear superalkali clusters C_3_O[C(CN)_2_]_2_M_3_ (where M = Li, Na, and K) optimized at CAM-B3LYP/6-311++g(d,p) are given in Figure 1. The studied polynuclear structures (**C1** to **C3**) show *C_2V_* point group symmetry (Table 1). These clusters are planar with a central carbon core. The determined bond distances between alkali metal and oxygen (d_-M-O_) are in increasing fashion with the increased size of metals (Li to K). The calculated d_-M-O_ bond distances for **C1**, **C2**, and **C3** are 1.82, 2.24, and 2.58 Å, respectively (Table 1). The observed geometric parameters (Appendix A) are very consistent with the previously reported study in the literature. Furthermore, these polynuclear clusters also show the increased bond distance between metals and nitrogen (d_M-N_). The observed bond lengths (d_M-N_) are 1.89, 2.26, and 2.62 Å for **C1, C2,** and **C3** clusters. The observed monotonic increase in bond lengths from Li to K may be attributed to increased metal size. The performed frequency calculation shows that there is no imaginary frequency associated with these clusters, and these are true minima on the potential energy surface.

The thermodynamic stability of the studied polynuclear clusters is evaluated through calculated binding energy per atom (E_b_). Overall, the binding energies range from −160.1 to −162.1 kcal mol^−1^ (Table 1), where the highest energy is found for **C1,** while the lowest is observed for the **C2** cluster. The obtained significant binding energies per atom suggests their thermodynamic stabilities. The calculated binding energies are higher in comparison to previously reported superalkali clusters NM’M (where M = Li, Na and K), C_3_X_3_Y_3_ (X = O, S, and Y = Li, Na and K) and bimetallic superalkali clusters [47,48]. The trend of binding energies per atom for studied clusters is also shown in Figure 2. Compared to clusters **C2** and **C3**, cluster **C1** has a greater binding energy value. The computed binding energies show high thermal stability of these clusters, which demonstrate that they can be synthesized experimentally.

### 2.2. Electronic Properties and Stability

The electronic stability and superalkali nature of these clusters can be observed from calculated ionization potential and electron affinity. The obtained vertical ionization potential values are smaller than Cs-atom (3.89 eV), which shows the superalkali characteristics of these clusters. These values are also significant and account for the electronic stabilities of these clusters. The highest VIE value of 3.65 eV is found for **C1**, while the lowest value (3.0 eV) is indicated for the **C3** cluster (Table 1). A gradual decrease in VIE values with the increased size of alkali metals can be seen in these clusters. On the other hand, the vertical electron affinity (VEA) values range from 0.27 to 0.89 eV, where **C3** shows the lowest value. The reduced values of EA indicate the electropositive nature of these clusters.

To obtain reactivity and charge distribution, the computed NBO charges are given in Table 1. The NBO charges (positive) on alkali metals slightly increase from Li to K metals. The charge is transferred from alkali metals to electronegative atoms (oxygen and nitrogen) within clusters. The NBO charges on alkali metals (QM) lie in the range of 0.58 to 0.62 e, where **C1** shows higher charge (positive magnitude) on metals. The charge transferred from alkali to O-atom is more pronounced as compared to the alkali to N-atom transition, which may be attributed to the higher electronegativity of the oxygen atom. The calculated NBO charges on O-atom (QO) lie in the range of −0.83 to −0.96 e and are higher for small-sized metals. 

### 2.3. Global Reactivity Descriptor

To characterize the reactivity of these clusters, we calculated global reactivity descriptor, chemical hardness, softness, and chemical potential (Table 2). The chemical hardness is measured as resistance to change in electronic distribution within clusters. The results obtained show that the **C3** cluster has the highest value (1.839 eV) of hardness, whereas the **C1** has the lowest value. The size of alkali metals is an obvious factor in controlling the hardness of clusters. The decreased values show a correlation with increased atomic size (Li to K), which guarantees soft nature and reactivity (Table 2). Similarly, the values of chemical softness (S) increase from **C1** to **C3** and reach the maximum of 0.33 eV.

The chemical potential values are also calculated and given in Table 2. The higher chemical potential (χ) values show the escaping tendency of the electrons in clusters and molecules. Obtained significant values (negative) indicate the stability of these polynuclear clusters. These values also suggest that the clusters do not decompose spontaneously into atoms and possess reasonable electronic stability. 

### 2.4. FMO Analysis and Excess Electron Nature of Clusters

To provide deep insight into the electronic structures of the studied clusters, the densities of the highest occupied molecular orbitals (HOMO) and virtual orbitals are plotted, and their energy values are given in Table 2. The HOMO and LUMO are quite important in quantum chemistry, as they allow the prediction of chemical stability and reactivity of molecules. Imperatively, the small difference between HOMO-LUMO (E_H-L_) is crucial for the description of reactivity of molecules. The smaller E_H-L_ gaps depict greater chemical reactivity with a high tendency to be polarized, as well as low kinetic stability. The HOMO-LUMO gap values lie in the range of 4.08 to 1.96 eV, where the highest value corresponds to **C1** clusters, while the lowest values correspond for **C3**. One can note that E_H-L_ decreases with increased metals size (Li to K) within clusters. Furthermore, decreased E_H-L_ gaps for the studied clusters can be attributed to increased energies of occupied orbitals where the energy of virtual orbitals goes on decreasing. 

The reactivity and conducting qualities of these clusters are revealed by a significant reduction in HOMO-LUMO gaps; there are excitable valence electrons (excess electrons) with transition HOMO → LUMO. The excess electron nature is further justified by the distribution of HOMO densities, and the electronic density cloud is mainly spread over alkali metals, which indicates the excess electron character of these superalkali clusters. The three-dimensional HOMO density of **C1** is shaped as a s-orbital, while for **C2** and **C3,** its look like a diffuse p-orbital (Figure 3). The LUMO densities that are generated are spherical and resemble s-orbitals.

### 2.5. Electrical Conductivity (σ)

The electrical conductivity is also a crucial aspect to demonstrate the NLO properties of molecules. The electrical conductivity (σ) is the function of energy gaps (E_H-L_); thus, narrowing HOMO-LUMO gaps more will lead to higher electrical conductivity of materials. In our designed clusters, the HOMO–LUMO gaps are significantly reduced from 4.08 to 1.96 eV. The electrical conductivity increases with increased size of alkali metals, which might be attributed to ease in excitation of electrons (HOMO to LUMO). 

### 2.6. TD-DFT Analysis

In the transparent region, the applications of nonlinear optical materials can be better understood. The obtained TD-DFT parameters of crucial transitions and first allowed transitions are given in Table 3. The percentage contribution of particular orbitals of these clusters for both transitions are also given in Table 3, whereas spectra are shown in Figure 4. The higher value of ϵ shows strong absorption at particular wavelength. Additionally, a higher value of *f_o_* reveals the strong transition probability. The studied cluster C3 has significant value of ϵ and oscillator strength at higher wavelength. The absorption maxima (λ_max_) during crucial transition for **C1, C2**, and **C3** are 758, 688, and 995 nm, respectively, where the redshifted (i.e., bathochromic sift) in λ_max_ is observed for **C3** (Table 3). The obtained excitation energies of crucial transition are 1.63, 0.92, and 1.24 eV for **C1**, **C2**, and **C3** clusters. On the other hand, the obtained optical gaps during allowed transitions are 1.63, 0.92, and 0.86 eV. The **C1** cluster has same value for crucial excitation and optical gap, while for **C2** and **C3**, optical gaps (allowed transition) values are significantly reduced. The excitation energies of allowed transition are decreasing monotonically from **C1** to **C3** with increased metal size (Li to K). The absorption maxima (λ_max_) of allowed transition are observed at longer wavelength as compared to absorption during crucial transition. As a result, bigger alkali metals have a stronger influence on absorptions shift to higher wavelengths. Furthermore, these clusters are completely transparent under the deep-UV region and have broadband absorption in the near-Visible region (Figure 4). The highest energy state TD-DFT parameters also reveal transparency in the deep-UV region, while absorption is mostly in the UV-visible region (Table 3). Likewise, the gradual increase in oscillator strength (*f*_o_) can also be seen for **C1** to **C3** clusters for crucial transition and allowed transition, which suggest increased quantum chemical excitation probabilities in higher-sized clusters.

### 2.7. Dipole Moment (µ_o_) and Change in Dipole Moment (Δµ)

For better comprehension of the electronic properties in these clusters, the dipole moment (µ_o_) and change in dipole moments (Δµ) values are also calculated. Overall, the dipole moment and change in dipole moment (Δµ) values are quite significant, which reveal asymmetric electronic distribution in these clusters (Table 4). The measured total dipole moment indicate polarity in clusters and the values of µ_o_ are significant and range from 1.49 to 4.11 au, where the highest value is observed for the **C3** cluster. On the other hand, the total change in dipole moment (Δµ) values are slightly smaller than those of dipole moment, but **C2** shows a significant value of 4.69 au (Table 4).

### 2.8. Linear and Nonlinear Optical (NLO) Properties

To investigate the influence of excess electrons on triggering the NLO properties of studied polynuclear clusters, hyperpolarizability (β_o_) and second hyperpolarizability (γ_o_) are two crucial evaluation indices. The presence of excess electrons greatly increases the hyperpolarizability and second hyperpolarizability values, as shown in a number of studies [43,47,48,49,50,51,52,53,54]. We are interested in exploring the role of excess electrons in decreasing excitation energies, which ultimately escalates hyperpolarizabilities. The calculated linear and NLO parameters for the C_3_O[C(CN)_2_]_2_M_3_ (where M = Li, Na and K) at CAM-B3LYP/6-311++g(d,p) clusters are given in Table 4. The α_o_ values lie in the range of 2.5 × 10^2^ to 6.62 × 10^2^, and there is a slight increase with the increased size of alkali metals. These values show liner optical properties of polynuclear clusters, and the presence of polarizabilities is due to asymmetric electronic density distribution in these clusters. 

The hyperpolarizability values of studied clusters range from 2.37 × 10^3^ to 5.78 × 10^3^ au, where the highest value is obtained for **C3,** while the lowest value is for the **C1** cluster. β_o_ values are increasing from Li to K metals within these clusters, which shows size dependence. It can be seen that electronic properties significantly contribute to hyperpolarizability response, and the larger the change in dipole moment, the higher the hyperpolarizabilities are. Thus, β_o_ values follow the increasing trend in these clusters, **C1** < **C2** < **C3**. Furthermore, the increased β_o_ values have a good match with reduced ionization potential and HOMO–LUMO gaps. The trend of size-dependent *β_o_* is also shown in Figure 5.

In addition, the static second hyperpolarizability (γ_o_) values are also calculated and lie in the range of 2.9 × 10^5^ to 5.5 × 10^6^ au (Table 4). Overall, γ_o_ values are significant where the highest value (5.5 × 10^6^ au) is obtained for the **C1** cluster, while the lowest is for **C3**. It is found that, with the increased size of alkali metals, the γ_o_ values decrease slightly from Li to Na and then dramatically for K. These values follow decreasing trend of γ_o_ values in order of **C1** > **C2** > **C3**. The calculated significant γ_o_ values guarantee the superior NLO properties of polynuclear clusters. The calculated values of β_o_ and γ_o_ are quite significant as compared to previously reported M_2_OCN superalkalis [47], M_2_X (where M = Li, Na and X = F, Cl) superalkali clusters, and lithium-based superalkalis Li_n_ (*n* = 3, 5, and 7) [55].

Furthermore, the β_vec_ values are strongly correlated with total hyperpolarizability (β_o_). The calculated β_vec_ values are given in Table 4. These values range from 2.37 × 10^3^ to 5.78 × 10^3^ au. The β_vec_ is the projection of hyperpolarizability on dipole moment vector and shows close resemblance β_o._ However, good agreement between β_o_ and β_vec_ shows that the direction of the dipole moment vector and the projection of hyperpolarizability are in the same direction. The factor affecting β_vec_ values might be the same for β_o_, where the highest β_vec_ values are obtained for higher-sized alkali metals (Table 4).

### 2.9. Scattering Hyperpolarizability (β_HRS_) 

Density functional theory calculations have been carried out to find scattering hyperpolarizability (β_HRS_), and values range from 1.34 × 10^3^ to 1.62 × 10^4^ au, where values are increasing steadily from the **C1** to **C3** cluster. The computed highest value is (1.62 × 10^4^ au), found for **C1** cluster, whereas the lowest value of 1.34 × 10^3^ au is for the **C1** cluster (Table 4). The β_HRS_ is the most viable parameter to calculate the hyperpolarizability of centrosymmetric molecules and clusters, even with zero change in dipole moment. There is an excellent agreement of β_HR*S*_ with β_o_ where the β_HRS_ show dependence on the size of alkali metals (M). The increased size of alkali metals (Li to K) favors the excellent electronic properties. Therefore, it also causes significantly enhanced *β_HRS_* values. Additionally, average dipolar and octupolar hyperpolarizability, which are more prominent in **C2** and **C3** clusters, provide a notable contribution to β_HRS_. Moreover, these clusters are of octupolar molecules, which can be seen by their highest octupolar contribution Φβ(j = 3) of 92 % for **C3** (Table 4). 

### 2.10. Frequency Dependent NLO Properties

We theoretically examined the incident–frequency (ω) effect on the first and second hyperpolarizability at applied frequencies of 532 and 1064 nm. The frequency-dependent first hyperpolarizability β(ω) is calculated with the electro–optical Pockel’s effect (EOPE) and second harmonic generation (SHG), whereas the γ(ω) is expressed in terms of dc-Kerr effect and second harmonic generation (SHG). Overall, the dynamic hyperpolarizabilities values are higher than those of static hyperpolarizabilities. The significant EOPE effect β(−ω; ω,0) was observed for the **C3** cluster at 532 nm, while its SHG value increased up to 1.7 × 10^6^ au (Table 5). It can be demonstrated that the dynamic hyperpolarizabilities are higher at the smaller incident frequency (ω = 532 nm) and slightly decreased at the higher dispersion frequency (1064 nm). Strikingly, the β(ω) values are much more pronounced for the EOPE effect at both frequencies.

Furthermore, the γ(ω) values are higher than γ_o,_ and the highest dc-Kerr value increased up to 2.6 × 10^9^ au for **C3** at 1064 nm (Table 6). The γ(ω) response becomes significant at the higher dispersion frequency (1064 nm), where SHG values are notable at both frequencies. The obtained higher values of the dc-Kerr effect reveal the nonlinear change in the refractive index of studied clusters. Hence, studied clusters have excellent NLO properties and can be used to design high-performance SHG devices. 

## 3. Computational Details

All density functional theory (DFT) calculations are performed in the gas phase with Gaussian 09 software, whereas visualization is achieved using the GaussView 5.0 program [56,57]. Geometries of all polynuclear C_3_O[C(CN)_2_]_2_M_3_ (where M = Li, Na, and K) clusters are optimized at CAM-B3LYP/6-311++G(d,p) functionality [58]. The quantum mechanics-based Coulomb attenuating method (CAM-B3LYP) is a hybrid exchange-correlation functional that combines B3LYP’s hybrid features with the CAM functional’s long-range corrected parameter. It was found that this long-range corrected density functional substantially reduces the overestimation seen with conventional techniques and typically provides results that are comparable to those of coupled cluster calculations. Previous research has demonstrated that this method is well recognized for examining molecules and clusters, as well as for determining NLO properties [59,60]. Besides, the choice of a suitable basis set is crucial for obtaining reliable results. Thus, the CAM-B3LYP method with 6-311+G(d,p) split valence basis set is a reliable level of theory for geometry optimization and accuracy in results for electronic properties [61,62,63,64,65].

To determine whether the presented structures are true minima on the potential energy surface, frequency calculations are carried out. For thermodynamic stability, we calculated binding energy per atom for these clusters. Electronic stability and superalkali nature are validated through computed ionization energies (IE) and electron affinities (EA). To further explore the electronic properties, we performed frontier molecular orbital (FMO) analysis. Natural bonding orbitals (NBO) study is carried out to explore the charge distribution on atoms within superalkali clusters [66]. The binding energy per atom (E_B_) is given by the following relations:(1)$$E_{B}=[E_{T}\left(X\right)-E_{A}{\left(X\right)}^{0}]/n\;$$
where E_T_ is the total electronic energy of studied (X) superalkali clusters, E_A_(X) is the total energy of individual atoms within clusters, and n is the total number of atoms. The vertical ionization energy, electron affinity, and electrical conductivity (σ) can be represented by the equation:VIE = E_X_^+^ − E_X_^0^(2)
VEA = E_X_^0^ − E_X_^−^(3)
(4)σ ∝ exp−EG2kT
where VIE and VEA are vertical ionization energies and electron affinities of studied clusters. In Equation (4), σ, E_G_, k, and T represent the electrical conductivity, energy gap, Boltzmann constant, and temperature, respectively. To further explore the electronic properties of studied clusters, we also performed total density of state (TDOS) analysis at the same method by using the GaussSum software [67]. The following equation can be used to explain the molecules under the static electric field.
(5)$$E\left(F\right)={\;E}^{0}-{\;µ}_{i}F_{i}-\frac{1}{2}\alpha_{\mathrm{ij}}F_{i}F_{j}-\frac{1}{6}\beta_{\mathrm{ijk}}F_{i}F_{j}F_{k}-\frac{1}{24}\gamma_{\mathrm{ijkl}}F_{i}F_{j}F_{k}F_{l}\ldots \;$$
where F is an external applied electric field, F_i_ is the component of field along i direction, E^0^ is the total energy of the superalkali clusters without a static electric field, and µ_i,_ α_ij_, β_ijk_, and γ_ijkl_ are dipole moment, polarizability, hyperpolarizability, and second-order hyperpolarizability, respectively. The mean dipole moment (µ_o_), change in dipole moment (Δµ), static polarizability (α_o_), and static first hyperpolarizability (β_o_) are calculated to illustrate the NLO response and associated responsible factors.

α_o_ = 1/3 (α_xx_ + α_yy_ + α_zz_)(6)β_o_ = (β_x_^2^ + β_y_^2^ + β_z_^2^)^1/2^(7)
where β_x_ = β_xxx_ + β_xyy_ + β_xzz_, β_y_ = β_yyy_ + β_yzz_ + β_yxx_ and β_z_ = β_zzz_ + β_zxx_ + β_zyy_.

µ_o_ = (µ_x_^2^ + µ_y_^2^ + µ_z_^2^) ^½^(8)

To obtain absorption behaviors and excited state parameters of studied clusters, we performed TD-DFT simulations. We considered 30 states for getting excited states parameters. The Gaussian band shape and the absorption spectra were obtained by using the following relation, ϑ:(9)$$\varepsilon_{o}\left(\bar{\vartheta }\right)=\varepsilon_{i}{}^{max}\;\exp\;\left[-{\left(\frac{\bar{\vartheta }-{\bar{\vartheta }}_{i}}{\sigma }\right)}^{2}\right]$$
where the *i* subscript represents the electronic excitation of interest. The other symbols in the equation have the following meanings:

${\bar{\vartheta }}_{i}$, shows the excitation energy (in wavenumbers) corresponding to the required electronic excitation in TD-DFT$\varepsilon_{i}{}^{max}$ is the value of at the maximum of the band shape Sigma (σ) is a wavenumber representation of the standard deviation that is related to the simulated band’s width. 

The second static hyperpolarizability *(*γ_o_*)* and the projection of hyperpolarizability on the dipole moment vector *(*β_vec_*)* are also calculated for our studied superalkali clusters at the same level of theory. Static second hyperpolarizability *(*γ_o_*)* and vector part of hyperpolarizability *(*β_vec_*)* are expressed as:
< γ > = 1/5 (γ_xxxx_ + γ_yyyy_ + γ_zzzz_ + γ_xxyy_ + γ_xxzz_ + γ_yyxx_ + γ_yyzz_ + γ_zzxx_)(10)
(11)βvec=∑µiβi|µ|

Moreover, the molecular parameters relevant to electro-optical Pockel’s effect (EOPE) and second harmonic generation (SHG) are calculated at externally applied frequencies (532 and 1064 nm).

## 4. Conclusions

In summary, we presented the geometric, electronic, and nonlinear optical properties of polynuclear carbon-based clusters at CAM-B3LYP/6-311++G(d,p) level. These clusters are thermodynamically stable, and their binding energies per atom range from −160.07 to −162.07 kcal mol^−1^. The electronic stability and superalkali nature are characterized through calculated ionization potential (IP) and FMO analyses. Small ionization potential further suggests their superalkali nature. NBO charge analysis reveals excellent charge separation within clusters. The performed DOS analysis shows size-dependent electronic and conductive properties, where **C3** is a potential candidate. The significant first and second hyperpolarizabilities, up to 5.78 × 10^3^ and 5.55 × 10^6^ au, respectively, are calculated for the clusters. The β_o_ response shows dependence on the size of alkali metals. Furthermore, the absorption study shows their transparency in the deep-UV region for NLO applications and absorption at longer wavelengths. The significant scattering hyperpolarizability (β_HRS_) value is (1.62 × 10^4^), calculated for the **C3** cluster, where octupolar contribution to β_HRS_ is 92%. The dynamic first hyperpolarizability β(ω) is more pronounced for the EOPE effect at 532 nm, whereas SHG is more prominent for second hyperpolarizability γ(ω).

## Figures and Tables

**Figure 1 molecules-28-01827-f001:**
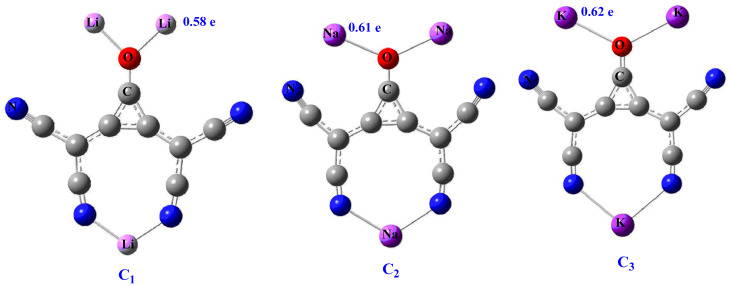
Optimized structure of Clusters **C1** to **C3**.

**Figure 2 molecules-28-01827-f002:**
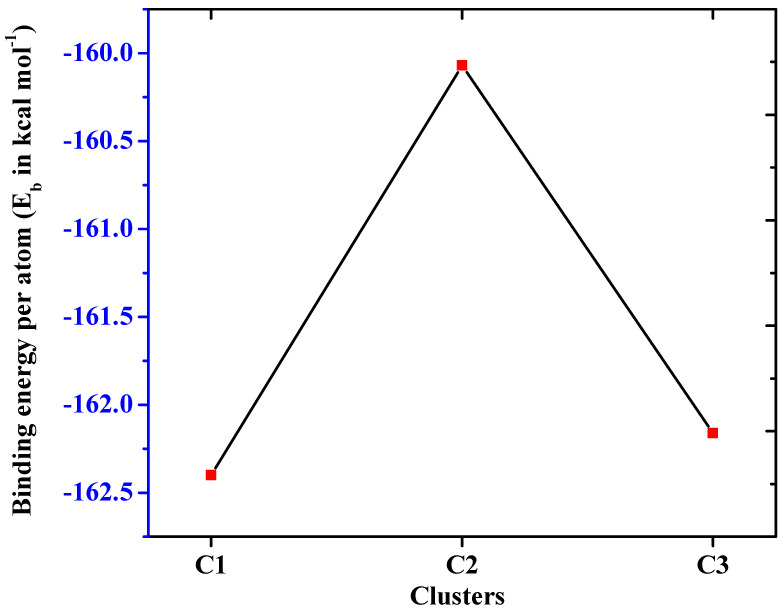
Binding energies per atom (E_b_) of clusters.

**Figure 3 molecules-28-01827-f003:**
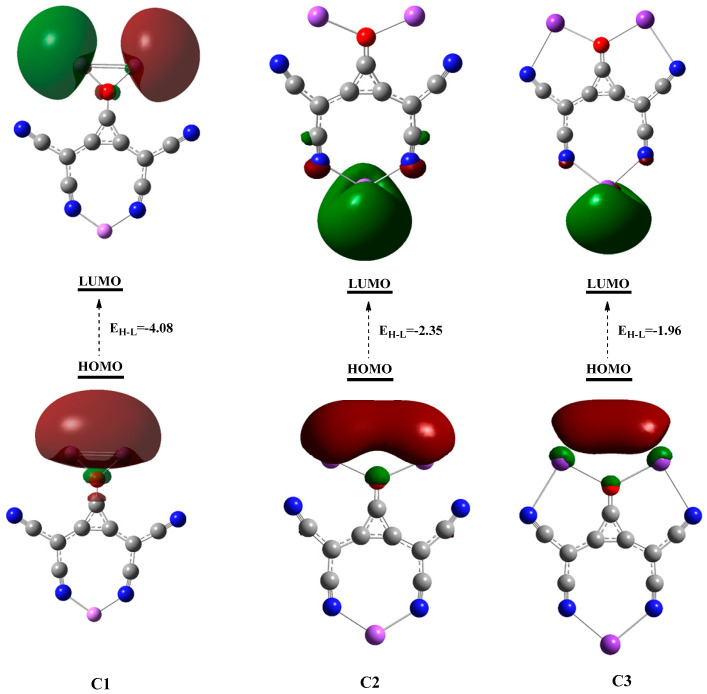
Generated HOMO and LUMO densities of clusters.

**Figure 4 molecules-28-01827-f004:**
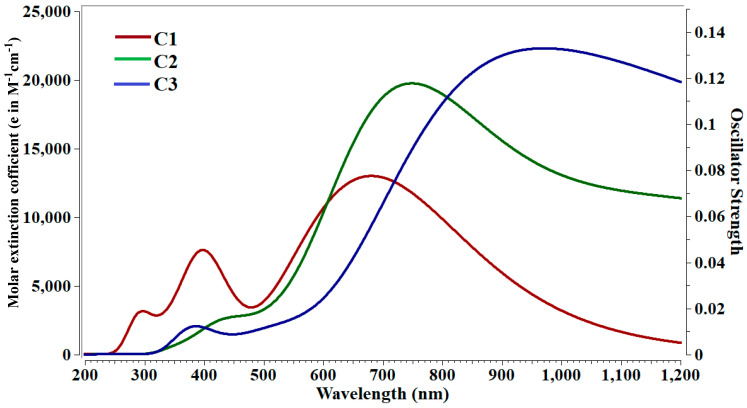
Absorption spectra of C1 to C3 clusters.

**Figure 5 molecules-28-01827-f005:**
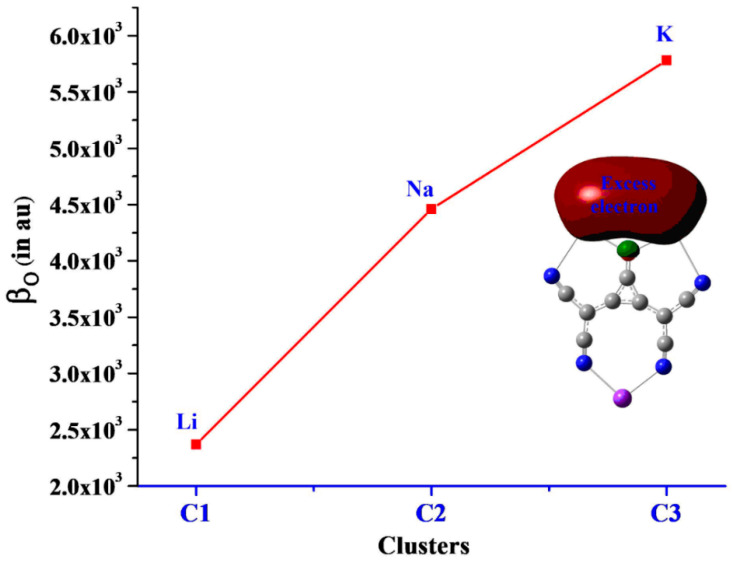
The representation of size-dependent hyperpolarizability (β_o_).

**Table 1 molecules-28-01827-t001:** Computed bond length between metal and O-atom (d_M-O_ in Å), the bond length between alkali metals and N-atom (d_M-N_ in Å), binding energies per atom (E_b_ in kcal mol^−1^), NBO charges on metals (QM), NBO charge on nitrogen (QN), NBO charge on oxygen atom (QO), VIE (in eV), and VEA (in eV), of **C1** to **C3** clusters.

Cluster	d_M-O_	d_M-N_	E_b_	Q(M)	Q(N)	Q(O)	VIE	VEA
**C_1_**	1.82	1.89	−162.4	0.58	−0.485	−0.960	3.65	0.76
**C_2_**	2.24	2.26	−160.1	0.61	−0.521	−0.839	3.41	0.89
**C_3_**	2.58	2.62	−162.1	0.62	−0.523	−0.830	3.00	0.27

**Table 2 molecules-28-01827-t002:** Energies of HOMO (E_HOMO_ in eV), LUMO energies (in eV), HOMO-LUMO gaps (E_H-L_ in eV), chemical hardness (*η* in eV), chemical, softness (*S* in eV), chemical potential (χ in eV), oscillator strength (*f_o_* in au), excitation energies (eV), and maximum absorption (in nm) of C1 to C3 clusters.

Cluster	E_HOMO_	E_LUMO_	E_H-L_	*η*	*S*	χ
**C1**	−4.70	−0.61	4.08	1.839	0.27	−1.81
**C2**	−3.24	−0.88	2.35	1.721	0.29	−1.72
**C3**	−2.80	−0.83	1.96	1.505	0.33	−1.50

**Table 3 molecules-28-01827-t003:** TD-DFT parameters of crucial excited sates, first allowed transitions, and highest state for C1 to C3 clusters.

Clusters	TD-DFT Parameters from Crucial Transitions			
	ΔE (eV)	λ_max_ (nm)	*f_o_* (au)	Major Orbital Contribution
**C1**	1.63	758	0.19	HOMO→LUMO+2 (82%)
**C2**	1.80	688	0.26	HOMO→LUMO+3 (36%)
**C3**	1.24	995	0.28	HOMO→LUMO+5 (67%)
**TD-DFT Parameters from First Allowed Transitions**				
**C1**	1.63	758	0.19	HOMO→LUMO+2 (82%)
**C2**	0.92	1338	0.23	HOMO→LUMO+1 (99%)
**C3**	0.86	1441	0.25	HOMO→LUMO+1 (96%)
**TD-DFT Parameters for Highest Energy States**				
**C1**	5.12	242	0.0018	
**C2**	4.25	291	0.0005	
**C3**	3.92	315	0.0002	

**Table 4 molecules-28-01827-t004:** Polarizabilities (α_o_ in au), hyperpolarizabilities (β_o_ in au), second hyperpolarizability (γ_o_ in au), scattering hyperpolarizability (β_HRS_ in au), vector part of hyperpolarizability (β_vec_ in au), average hyperpolarizability (<β_J=1_> in au), average octupolar hyperpolarizability (<β_J=3_> in au), % dipolar contribution to hyperpolarizability Φβ(j = 1), and % octupolar contribution to hyperpolarizability Φβ(j = 3) of C1 to C3 clusters.

Clusters	α_o_	β_o_	γ_o_	β_HRS_	β_vec_	<β_J=1_>	<β_J=3_>	Φβ(j = 1)	Φβ(j = 3)
**C1**	2.5 × 10^2^	2.37 × 10^3^	5.5 × 10^6^	1.34 × 10^3^	2.37 × 10^3^	1.8 × 10^3^	3.37 × 10^3^	35%	65%
**C2**	5.07 × 10^2^	4.46 × 10^3^	1.2 × 10^6^	4.87 × 10^3^	4.46 × 10^3^	3.28 × 10^3^	1.49 × 10^4^	18%	82%
**C3**	6.62 × 10^2^	5.78 × 10^3^	2.9 × 10^5^	1.62 × 10^4^	5.78 × 10^3^	4.30 × 10^3^	5.20 × 10^3^	08%	92%

**Table 5 molecules-28-01827-t005:** Hyperpolarizability (β_0_ in au), frequency-dependent hyperpolarizability β(ω) in terms of electro-optic-Pockel’s effect (EOPE) β (−ω; ω, 0) in au, and electric field induced second harmonic generation (EFSHG) β (−2ω; ω, ω) in au at ω = 532 au.

Cluster	ω = 0	ω = 532 nm		ω = 1064 nm	
	β (0;0,0)	β (−ω; ω,0)	β (2-ω;ω,ω)	β (−ω; ω,0)	β (−2ω; ω, ω)
**C1**	2.5 × 10^2^	8.1 ×10^3^	2.2 × 10^5^	8.1 × 10^5^	2.9 × 10^5^
**C2**	5.0 × 10^2^	1.0 × 10^5^	4.2 × 10^5^	2.7 × 10^6^	1.6 × 10^5^
**C3**	6.6 × 10^2^	1.2 × 10^7^	1.7 × 10^6^	5.0 × 10^5^	4.5 × 10^3^

**Table 6 molecules-28-01827-t006:** Static second hyperpolarizability (γ_o_ in au), frequency-dependent second-hyperpolarizability γ(ω) in term of electro-optic-pockel’s effect (EOPE) γ (−ω; ω, 0) in au, and electric field-induced second harmonic generation (efshg) γ (2-ω; ω, ω) in au at ω = 532 au.

Clusters	ω = 0	ω = 532 nm		ω = 1064 nm	
	γ (0;0,0,0)	γ (−ω; ω,0,0)	γ (−2ω;ω,ω, ω)	γ (−ω; ω,0,0)	γ (−2ω; ω, ω, ω)
**C1**	5.5 × 10^6^	3.0 × 10^8^	6.0 × 10^7^	1.0 × 10^6^	1.2 × 10^8^
**C2**	1.2 × 10^6^	2.4 × 10^8^	2.3 × 10^7^	2.3 × 10^7^	5.0 × 10^7^
**C3**	2.9 × 10^5^	4.9 × 10^7^	1.7 × 10^8^	2.6 × 10^9^	2.0 × 10^9^

## Data Availability

The author confirms that data supporting finding current study are available within article and in its supporting information. Raw data that supports the finding of this study are available from the corresponding author’s upon request.

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
