# Peer review of "Geometric, Electronic, and Optoelectronic Properties of Carbon-Based Polynuclear C3O[C(CN)2]2M3 (where M = Li, Na, and K) Clusters: A DFT Study"

_molecules, 2023, doi:10.3390/molecules28041827_

Round 1

Reviewer 1 Report

The paper by Bayach et al. although of some potential interest, does require significant amendments and improvements both in the quality of presentation and from a methodological point of view.

Suggestions for improving the quality of presentation:

1) In the introduction there are way too many repetition of the same concept: introduction of excess electrons into molecules and clusters increases the hyperpolarizability. This is quite a trivial concept, which should be put in a more quantitative basis, and does not warrant reiteration in the text. Moreover, what does "excess model" means? (end of 2nd paragraph of the introduction) . The language should be corrected for misspellings and the quality is rather poor. It should be shortened, and completely rewritten. More importantly the authors should clearly explain why they chose those particular compounds to study, and contrast them with similar systems in the current literature.

2) In the computational details there are plenty of misspellings ("tripple Zetta", "sueralkali" etc. The language is poor, for example "The system under the applied electric field is given by the equation:" CAM-B3LYP is a range-separated xc functional, not a function.

3) Figure 2 should be improved (the vertical axis is missing for example, moreover it does not add to the discussion, since there are only 3 points to display).

4) What is the definition of "Epsilon" in Figure 3? Should the authors explain its relation to oscillator strength as obtained from a standard TDDFT calculation?

5)Figure 5 should be moved to a Supporting Information file with its discussion since it does not add any new insight.

Suggestions for improving the soundness of the work

1)Why are the VIE negative (Table 1) ?? Either it is a trivial mistake, or there is something really wrong with the calculations. Check also the values of VEA, from their definition of Eq. (4), it is suggested that the negative ions are unstable. Moreover, NBO charges on the alkali metals are very similar for clusters C1-C3, contrary to what is written in the text.

2) The authors should comment on the reliability of HOMO-LUMO gaps provided by the DFT calculations, and relate it to the optical gap value obtained from the TDDFT calculations

3) The HOMO densities do not resemble an s orbital in Figure 3. Why should they??

4) How many roots have been requested in the TDDFT calculations? The authors claim that C1-C3 are transparent in the UV region, which is clearly a wrong statement by looking at the profiles displayed in Figure 4. However, the authors should check that the number of eigenvectors of the Omega matrix which they extract is sufficient to cover the excitation energy range of interest. No mention of this is given in the computational detail section. Does the excitation energy \Delta E reported in Table 2 refer to the first optically allowed transitions? Is it the optical gap?

5) Since the results obtained with the two-level model are so different from the computed TDDFT values of hyperpolarizability, the analysis of the controlling factors is fundamentally flowed, and, in my opinion, useless.

In summary, I think the paper, in is current form, is, without any doubt, unsuitable for publication in Molecules. A *very* extensive and fundamental revision of the manuscript is needed.  My opinion is that the paper should be rejected.

Author Response

Reviewer 1

Suggestions for improving the quality of presentation:

1) In the introduction there are way too many repetition of the same concept: introduction of excess electrons into molecules and clusters increases the hyperpolarizability. This is quite a trivial concept, which should be put in a more quantitative basis, and does not warrant reiteration in the text. Moreover, what does "excess model" means? (end of 2nd paragraph of the introduction) . The language should be corrected for misspellings and the quality is rather poor. It should be shortened, and completely rewritten. More importantly the authors should clearly explain why they chose those particular compounds to study, and contrast them with similar systems in the current literature.

Answer: We are thankful for the valuable comment of reviewer. We have removed the repeated sentence from introduction section. We have polished the excess electron concept in more comprehensive manners. All the misspellings and technical mistakes are removed. Furthermore, we have shortened the introduction. Excess model was a mistake and it has been removed. We have now added the reason for choosing these systems at the end of the Introduction (in the last paragraph). The results in this study are compared with those in the current literature in the results and discussion section.

2) In the computational details there are plenty of misspellings ("tripple Zetta", "sueralkali" etc. The language is poor, for example "The system under the applied electric field is given by the equation:" CAM-B3LYP is a range-separated xc functional, not a function.

Answer: We have taken care of all mistakes related to grammar, spelling etc.

3) Figure 2 should be improved (the vertical axis is missing for example, moreover it does not add to the discussion, since there are only 3 points to display).

Answer: We have improved the Figure 2. We have also added discussion related to this figure.

4) What is the definition of "Epsilon" in Figure 3? Should the authors explain its relation to oscillator strength as obtained from a standard TDDFT calculation?

Answer: We have now given expression for the calculations of epsilon from oscillator strength in Computational Methodology. Oscillator strength is obtained directly from TD-DFT calculations.

5)Figure 5 should be moved to a Supporting Information file with its discussion since it does not add any new insight.

Answer: We have moved the Figure 5 to supporting information along with its discussion.

Suggestions for improving the soundness of the work

1)Why are the VIE negative (Table 1) ?? Either it is a trivial mistake, or there is something really wrong with the calculations. Check also the values of VEA, from their definition of Eq. (4), it is suggested that the negative ions are unstable. Moreover, NBO charges on the alkali metals are very similar for clusters C1-C3, contrary to what is written in the text.

Answer: We are sorry for the mistakes. VIE were incorrectly presented (a trivial mistake).The negative sign is now removed from VIE values and these value are rechecked according to equation 4 (see Table 1). Also, we have corrected the VEA values in Table 1. In our original draft, VEA were not converted from atomic units into eV. We have now converted these values. The presented data are carefully rechecked, and we believe that the given values are now correct. The discussion of NBO charges is also corrected.

2) The authors should comment on the reliability of HOMO-LUMO gaps provided by the DFT calculations, and relate it to the optical gap value obtained from the TDDFT calculations

Answer: We have calculated optical gaps value using TD-DFT simulations and their comparison to HOMO-LUMO gaps from DFT is also given (see Table 3). Our results reveals that the optical gap values are quite smaller than those of HOMO-LUMO gaps.  

3) The HOMO densities do not resemble an s orbital in Figure 3. Why should they??

Answer: We have corrected the discussion about shape of electronic density. The C1 cluster has shape like s-orbital while of C2 and C3 are not s type rather they are more scattered. The LUMO densities resembles to p-orbitals.

4) How many roots have been requested in the TDDFT calculations? The authors claim that C1-C3 are transparent in the UV region, which is clearly a wrong statement by looking at the profiles displayed in Figure 4. However, the authors should check that the number of eigenvectors of the Omega matrix which they extract is sufficient to cover the excitation energy range of interest. No mention of this is given in the computational detail section. Does the excitation energy \Delta E reported in Table 2 refer to the first optically allowed transitions? Is it the optical gap?

Answer: We had considered 20 states for TDDFT simulations in our original calculations. But, in response to the reviewer comment, we have recalculated UV-Vis spectra with 30 states. The transparency of clusters can be seen in deep-UV region (0-20 nm) and have absorption in UV-visible region. We also have corrected the statement about the transparency of these clusters (see page 13). The excitation energies (ΔE) reported in table 2 are crucial excitation energies (with highest oscillator strengths). These energies are not corresponding to optical gap. Optical gap values are now added in Table 3

5) Since the results obtained with the two-level model are so different from the computed TDDFT values of hyperpolarizability, the analysis of the controlling factors is fundamentally flowed, and, in my opinion, useless.

Answer: values of hyperpolarizability obtained from two-level model show the same trend as for βo. However, agreeing to the reviewer, discussion on controlling factor is removed from the revised manuscript

In summary, I think the paper, in is current form, is, without any doubt, unsuitable for publication in Molecules. A *very* extensive and fundamental revision of the manuscript is needed.  My opinion is that the paper should be rejected.

Reviewer 2 Report

The paper by Bayach et al. presents the results of DFT calculations of geometrical, electronic and NLO-related properties of some alkali metal-containing carbon-based clusters. The study is purely theoretical: there are no indications that such systems were (or even could be) synthesised and therefore there are no relevant experimental results. Nevertheless, the findings may be of interest in the context of NLO properties of small clusters, therefore the paper could be accepted after some errors are corrected and some improvements are made.

There are some issues to be considered in the revision:

1. The sign in equation 3 is wrong. By definition, electron affinity is the amount of energy *released* when an electron is being attached to the neutral system (see the IUPAC gold book
https://doi.org/10.1351/goldbook.E01977 ). Equation 3 gives therefore the -EA value, instead of EA. Accordingly, a positive value of EA indicates that the electron bounds to the system. The statement "The positive values of EA indicate that electrons are not bound with metals (loosely bound)." in sec. 3.2 is therefore wrong.
Eq. 3, calculated values of EA and the discussion have to be corrected. Fortunately, as the calculated EA values are close to zero, the conclusions will not be much affected.

2. Sec. 3.6: "It can be seen that these clusters are completely transparent under the UV region". Apparently, this is not true, as the absorption in the near-UV between 300-400 nm is visible in Figure 4.
It was not stated in Sec. 2 how many excited states were calculated in TD-DFT. My guess is that only 3 states (default value used by Gaussian) and therefore the analysis of the low-wavelength part of the spectrum is incomplete.
To obtain properly the spectrum in the UV and far-UV region, one has to appropriately increase the number of states calculated in TD-DFT.

3. How the spectra shown in Figure 4 were obtained from the calculated data?
It would be beneficial to the discussion to show in the table the calculated transition energies, oscillator strengths and the parentage of the transitions (occupied-virtual orbital pairs contributing mostly to the transition). What is the parentage of the most intense, low-energy transition (HOMO-LUMO, other pair of orbitals)?
I guess that C1 will differ from C2 and C3 with this respect, because of the difference in LUMO.

4. Authors should be more careful with the wording. Sec. 2: "we mainly calculated the electro-optical Pockel’s effect (EOPE), and second harmonic generation (SHG) phenomena".
Actually, only the molecular parameters relevant to EOPE or SHG were calculated, not the EOPE or SHG signals which could be measured in the experiments.

5. Some technical issues:
- Eq. 1: use of "nE" for the total energy of separated atoms is rather confusing, because it suggests that this is the product n*E, where E is an energy of an individual atom (and this is not true, because atoms are different, so that there is no common E value)
- page 3: should be "triple zeta" not "tripple Zetta"
- table 1: wrong headings for bond distances
- sec. 3.2: there is no need to use modulus for e value (|e|), because e is positive itself. "e" stands for elementary charge and not for electron charge (electron charge is -1 e ).

Author Response

Reviewer 2

  1. The sign in equation 3 is wrong. By definition, electron affinity is the amount of energy *released* when an electron is being attached to the neutral system (see the IUPAC gold book
    https://doi.org/10.1351/goldbook.E01977 ). Equation 3 gives therefore the -EA value, instead of EA. Accordingly, a positive value of EA indicates that the electron bounds to the system. The statement "The positive values of EA indicate that electrons are not bound with metals (loosely bound)." in sec. 3.2 is therefore wrong.
    Eq. 3, calculated values of EA and the discussion have to be corrected. Fortunately, as the calculated EA values are close to zero, the conclusions will not be much affected.

Answer: We have corrected the equation 3 and the obtained values are now corrected (see Table 1). The more negative EA values favors the electron addition process.  We have corrected the statement in discussion (see page 8)

  1. Sec. 3.6: "It can be seen that these clusters are completely transparent under the UV region". Apparently, this is not true, as the absorption in the near-UV between 300-400 nm is visible in Figure 4.

Answer: we are sorry for incorrect statement. The transparency of clusters lies in deep-UV region (0-200 nm). These cluster show absorption in UV region. The statement is now corrected (see page. No.13)
It was not stated in Sec. 2 how many excited states were calculated in TD-DFT. My guess is that only 3 states (default value used by Gaussian) and therefore the analysis of the low-wavelength part of the spectrum is incomplete.
To obtain properly the spectrum in the UV and far-UV region, one has to appropriately increase the number of states calculated in TD-DFT.

Answer: We have rechecked the number of states in TD-DFT analysis. We considered 20 states for obtaining UV-Vis spectra in our original calculations. At the stage of revision, we have recalculated UV-Vis spectra with 30 states. Additionally, we have mentioned it in computational methodology section (see page 6).

3. How the spectra shown in Figure 4 were obtained from the calculated data?
It would be beneficial to the discussion to show in the table the calculated transition energies, oscillator strengths and the parentage of the transitions (occupied-virtual orbital pairs contributing mostly to the transition). What is the parentage of the most intense, low-energy transition (HOMO-LUMO, other pair of orbitals)?
I guess that C1 will differ from C2 and C3 with this respect, because of the difference in LUMO.

Answer: We have plotted UV spectra by TD-DFT study and their corresponding data are given in Table 3 (see page No 12,13). We have given all these required parameters for the crucial transition (transition with the highest oscillator strength) and the first allowed transition

4. Authors should be more careful with the wording. Sec. 2: "we mainly calculated the electro-optical Pockel’s effect (EOPE), and second harmonic generation (SHG) phenomena".
Actually, only the molecular parameters relevant to EOPE or SHG were calculated, not the EOPE or SHG signals which could be measured in the experiments.

Answer: We have improved the discussion for EOPE and SHG parameters in computational detail and discussion (see page 7).

5. Some technical issues:
- Eq. 1: use of "nE" for the total energy of separated atoms is rather confusing, because it suggests that this is the product n*E, where E is an energy of an individual atom (and this is not true, because atoms are different, so that there is no common E value)
- page 3: should be "triple zeta" not "tripple Zetta"
- table 1: wrong headings for bond distances
- sec. 3.2: there is no need to use modulus for e value (|e|), because e is positive itself. "e" stands for elementary charge and not for electron charge (electron charge is -1 e ).

Answer: We have corrected all technical issues

Reviewer 3 Report

The article “Geometric, electronic, and optoelectronic properties of carbon-based polynuclear C3O[C(CN)2]2M3 (where M=Li, Na, and K) clusters; A DFT study” investigates the electronic structure aspects of carbon polynuclear clusters. The manuscript is organized and presented well. The investigation may find broad interest among the researchers. The manuscript can be accepted for publication after minor corrections/revisions.

1.     What is the rationale for choosing CAM-B3LYP functional for the investigations?

2.     Does dispersion corrections influence the trend of the results? Why did authors not consider dispersion corrections?

Author Response

Reviewers 3

The investigation may find broad interest among the researchers. The manuscript can be accepted for publication after minor corrections/revisions.

  1. What is the rationale for choosing CAM-B3LYP functional for the investigations?

Answer: The choice of method is based upon previous similar studied where the optical and NLO parameters are calculated at same functional. It was found that this long-range corrected density functional removes to large parts the overestimation observed for standard methods and in many cases provides results close to those of coupled cluster calculations. Moreover, the results obtained here can be compared with the previous studies where this functional had been applied (see page 5 and NLO section).

  1. Does dispersion corrections influence the trend of the results? Why did authors not consider dispersion corrections?

Answer: Literature reveals that NLO properties are dependent on long range correction but not dispersion corrections. We did not come across any study in the literature where role of dispersion on nonlinear optical properties is discussed. We have adopted coulomb attenuating method (CAM-B3LYP) which is a hybrid exchange-correlation functional and it combines B3LYP's hybrid features with the CAM functional's long-range corrected parameter. Thus long-range corrected functional can captures correlation and their results are comparable to experimental data 

Round 2

Reviewer 1 Report

The author's have taken adequate steps to improve the quality of the research. I think that the paper can be published now.

Author Response

We are thankful to reviewer to recommend our article for publication into Molecules 

Reviewer 2 Report

The paper has improved in the revision. Few remaining issues can be corrected in a minor revision.

1. Provide information on the energy/wavelength of the highest state calculated in TD DFT, to confirm that this state is above the energies corresponding to the deep-UV region. Otherwise, one can not draw conclusions about the transparency in deep-UV.

2. Table 3 and Sec. 3.6 - what is the "crucial" transition? For C2 and C3 its oscillator strength is only little larger than that of the lowest allowed transition. How the "crucial" transition is defined then?

3. Lines 163-164: what was the value of sigma used to plot the spectra?

4. In the revision "|e|" was changed to "-1e"  (Sec. 3.2 and the caption of Table 1), and this is obviously wrong (whereas the original wording was just only awkward).
I repeat: "e" stands for elementary charge (and not the electron charge) and it is positive (in other words, e is the proton charge).
Therefore the correct way of presenting charges is "0.58 to 0.62 e" (lines 217-218); "-0.83 to -0.96 e"  (line 221) and "QM in e" (Table 1).

5. Headers "XM-O" and "XM-N" in Table 1 have not been corrected yet.

Author Response

  1. Provide information on the energy/wavelength of the highest state calculated in TD DFT, to confirm that this state is above the energies corresponding to the deep-UV region. Otherwise, one can not draw conclusions about the transparency in deep-UV.

Answer: We have reported the TD-DFT parameters for highest state in UV-Vis analysis. The absorption wavelengths are above deep-UV region (200 nm) which suggest their transparency.

2. Table 3 and Sec. 3.6 - what is the "crucial" transition? For C2 and C3 its oscillator strength is only little larger than that of the lowest allowed transition. How the "crucial" transition is defined then?

Answer: We have chosen crucial states on basis of oscillator strength values. Due to the slightly higher oscillator’s values we considered them as crucial transitions for TD-DFT analysis. Also, oscillator strength values are quite higher than those of highest states in UV-Vis analysis.

3. Lines 163-164: what was the value of sigma used to plot the spectra?

Answer: The σ is the standard deviation in wavenumbers, which is related to the width of the simulated band. Specifically, it is the half-width of the Gaussian band at ε=εmax/e.  The Sigma is also related to the half-width at half height of the gaussian band (HWHM) and "e" is the irrational number e (2.7182818285...), that is the number whose natural logarithm is 1 (ln e = 1) The relation of sigma to HWHM can  be written as

HWHM = sigma * sqrt(ln(2)) = sigma * 0.8325546112

For the plotted UV-Vis spectra, the default value of sigma used in GaussView is sigma=0.4 eV, which is equivalent to HWHM=0.333 eV or 2685.83 cm-1
4. In the revision "|e|" was changed to "-1e"  (Sec. 3.2 and the caption of Table 1), and this is obviously wrong (whereas the original wording was just only awkward).

Answer: Corrected
I repeat: "e" stands for elementary charge (and not the electron charge) and it is positive (in other words, e is the proton charge).
Therefore the correct way of presenting charges is "0.58 to 0.62 e" (lines 217-218); "-0.83 to -0.96 e"  (line 221) and "QM in e" (Table 1).

Answer: We have corrected the NBO charges presentation in discussion

5. Headers "XM-O" and "XM-N" in Table 1 have not been corrected yet.

Answer: Corrected